# ZNF281 Promotes Colon Fibroblast Activation in TGFβ1-Induced Gut Fibrosis

**DOI:** 10.3390/ijms231810261

**Published:** 2022-09-06

**Authors:** Ilaria Laudadio, Alex Bastianelli, Valerio Fulci, Claudia Carissimi, Eleonora Colantoni, Francesca Palone, Roberta Vitali, Elisa Lorefice, Salvatore Cucchiara, Anna Negroni, Laura Stronati

**Affiliations:** 1Department of Molecular Medicine, Sapienza University, 00161 Rome, Italy; 2Division of Health Protection Technologies, ENEA, 00123 Rome, Italy; 3Department of Maternal Infantile and Urological Sciences, Sapienza University, 00161 Rome, Italy

**Keywords:** intestinal fibrosis, Crohn’s disease, fibroblasts, ZNF281

## Abstract

Crohn’s disease (CD) and ulcerative colitis (UC) are chronic inflammatory disorders of the gastrointestinal tract. Chronic inflammation is the main factor leading to intestinal fibrosis, resulting in recurrent stenosis, especially in CD patients. Currently, the underlying molecular mechanisms of fibrosis are still unclear. ZNF281 is a zinc-finger transcriptional regulator that has been characterized as an epithelial-to-mesenchymal transition (EMT)-inducing transcription factor, suggesting its involvement in the regulation of pluripotency, stemness, and cancer. The aim of this study is to investigate in vivo and in vitro the role of ZNF281 in intestinal fibrogenesis. Intestinal fibrosis was studied in vivo in C57BL/6J mice with chronic colitis induced by two or three cycles of administration of dextran sulfate sodium (DSS). The contribution of ZNF281 to gut fibrosis was studied in vitro in the human colon fibroblast cell line CCD-18Co, activated by the pro-fibrotic cytokine TGFβ1. ZNF281 was downregulated by siRNA transfection, and RNA-sequencing was performed to identify genes regulated by TGFβ1 in activated colon fibroblasts via ZNF281. Results showed a marked increase of ZNF281 in in vivo murine fibrotic colon as well as in in vitro human colon fibroblasts activated by TGFβ1. Moreover, abrogation of ZNF281 in TGFβ1-treated fibroblasts affected the expression of genes belonging to specific pathways linked to fibroblast activation and differentiation into myofibroblasts. We demonstrated that ZNF281 is a key regulator of colon fibroblast activation and myofibroblast differentiation upon fibrotic stimuli by transcriptionally controlling extracellular matrix (ECM) composition, remodeling, and cell contraction, highlighting a new role in the onset and progression of gut fibrosis.

## 1. Introduction

Inflammatory bowel disease (IBD), subdivided into the two major entities Crohn’s disease (CD) and ulcerative colitis (UC), is a idiopathic, heterogeneous disorder characterized by a relapsing and remitting disease course due to an inappropriate immune response toward the intestinal microbiota in genetically predisposed individuals [1,2].

Epithelial barrier injury and tissue damage cause a disproportionate immune response, leading to chronic inflammation and, especially in CD patients, to excessive extracellular matrix (ECM) deposition and gut fibrosis [3]. Indeed, more than 50% of patients with CD develop a penetrating or stricturing course of disease due to fibrostenosis, of whom one-third will undergo surgical intervention [4].

Currently, the molecular mechanisms regulating fibrosis remain incompletely understood, and established targeted therapeutic strategies are still unsettled. However, most evidence indicates that gastrointestinal tract fibrosis occurs only in previously or actively inflamed regions, implying that persisting inflammation is a necessary condition for developing fibrosis. In physiological conditions, acute inflammation is normally followed by healing with tissue restoration and functional recovery; in pathological conditions, the fibrotic process is established by the persistence of the inflammatory stimulus [4,5,6,7,8,9]. Many studies have demonstrated that TGF-β stimulates the activation and proliferation of fibroblasts, resulting in extracellular matrix deposition and fibrosis in several organs, including lungs, kidneys, liver, skin, and gut [9,10,11,12,13,14]. Actually, inflammation causes the release of TGFβ, with the consequent activation of fibroblasts that differentiate into myofibroblasts, a cell type that has an intermediate phenotype between smooth muscle cells and fibroblasts. Myofibroblasts possess both contractile and migratory abilities and are responsible for the synthesis of ECM proteins [5].

Zinc-finger proteins (ZNFs) are one of the most abundant groups of proteins that have a key role in the regulation of important cellular processes. Alterations in ZNFs are involved in the development of several human diseases and in the progression of cancer, including colorectal cancer [15,16]. In particular, ZNF281, also known as ZBP-99 or ZNP-99, is a 99 kDa Krüppel-type zinc-finger transcriptional regulator that binds to GC-rich regions located in the promoters of a variety of genes, among which ornithine decarboxylase was the first target identified [17]. ZNF281 is expressed at high levels in the placenta, kidney, brain, heart, liver, and lymphocytes, whereas most other tissues display detectable albeit low ZNF281 expression [18]. ZNF281 has been characterized as an epithelial-to-mesenchymal transition (EMT)-inducing transcription factor, suggesting its involvement in the regulation of pluripotency, stemness, and cancer [19,20]. More recently, ZNF281 has emerged as a nexus of inflammatory gene programs [16]. Further, ZNF-281 has shown to be actively involved in intestinal inflammation controlling inflammatory genes, such as IL-8, IL-17, IL-23, and IL-1β. ZNF281 has also been shown to be significantly increased in the inflamed colon of patients with inflammatory bowel disease (IBD), and preliminary data have suggested its potential implication in gut fibrogenesis; however, the results are still limited and not exhaustive [21].

Here, we report findings that implicate, for the first time, ZNF281 as a direct participant in intestinal fibrogenesis. Following fibrosis induction, ZNF281 levels were significantly increased in murine fibrotic colon tissues. Further, the silencing of ZNF281 in activated colon fibroblasts showed that ZNF281 was required to activate the expression of fibrotic genes, leading to the activation of colon fibroblasts and their differentiation into myofibroblasts. Overall, our data identify ZNF281 as a new player in colon fibroblast activation upon fibrotic stimuli and provide important new insight into the molecular events that arbitrate intestinal fibrosis, opening new potential perspectives to its management.

## 2. Results

### 2.1. Zfp281/ZNF281 mRNA and Protein Expressions Are Significantly Increased from the Onset of Intestinal Fibrosis and Continue to Increase According to Its Progression

Inflammatory-derived gut fibrosis was induced by consequent treatments with two or three cycles of 2.5% DSS for 1 week, followed by 2 weeks of recovery [22], in C57BL/6J mice. Animals were sacrificed at 6 and 9 weeks to develop early and late fibrosis, respectively. The occurrence of fibrosis was observed by analyzing macroscopic (colon length and weight, animal weight), microscopic (histology, collagen fiber deposition), and molecular (Il-6, Tnfα, TGFβ1, Col3A1, Mmp9, Snail, αSma, fibronectin (Fn1), vimentin (Vim), and E-cadherin mRNA expression) parameters (Appendix A). The induction of murine early and late fibrosis, featured by an upward accumulation of ECM and an improved production of collagen, indicating the occurrence of progressive and increasingly severe colon fibrosis, served to highlight the role of ZNF281 in the development and progression of the fibrotic process. Results showed that Zfp281 (the murine orthologous gene of human ZNF281) mRNA expression was significantly increased after 6 and 9 weeks of treatment (Figure 1A). Coherently, the Zfp281 protein level was strongly upregulated already at the onset of fibrosis and increased in parallel with the progress of the severity of fibrosis for up to 9 weeks, as observed by Western blot and immunohistochemistry (Figure 1B,C and Appendix A). This finding demonstrates that Zfp281 is altered from the very early onset of intestinal fibrosis, and its expression levels correlate with the increasing severity of fibrosis.

### 2.2. ZNF281 Silencing Significantly Decreases Fibrotic Genes αSMA, SNAIL, and FN1 in Human Colon Fibroblasts

To investigate the relationship between ZNF281 and gut fibrosis, we used CCD-18Co cells, namely, human fibroblasts isolated from normal colon tissue, since fibroblasts are the main effector cells in the development of intestinal fibrosis. CCD-18Co were exposed to TGFβ1 for 24 h to trigger fibrosis, which was confirmed by the significant increase of the pro-fibrotic gene expression of αSMA, SNAIL, FN1, and fibroblast activation protein alpha (FAP) (Figure 2A). Likewise, ZNF281 mRNA and protein levels were significantly increased (Figure 2B).

Further, a specific siRNA to knock-down ZNF281 was used. Thus, CCD-18Co cells were transfected with a negative control siRNA (siCtl) and siZNF281 and exposed to TGFβ1 (hereinafter indicated as siCtl TGFβ1 and siZNF281 TGFβ1). Next, ZNF281 mRNA and protein expressions were analyzed to confirm gene downregulation (Figure 3A,B). Interestingly, results showed that upon ZNF281 silencing, TGFβ1-mediated stimulation of αSMA, SNAIL, and FN1 mRNA expressions was partially or completely abolished. It is worth noting that the downregulation of ZNF281 in fibroblasts not activated by TGFβ1 resulted in a reduction of pro-fibrotic gene expression (Figure 3C). Overall, these data suggest that ZNF281 is able to control colon fibroblast activation upon fibrotic stimuli.

### 2.3. ZNF281 Controls the Expression of Genes Involved in Cell Migration, Differentiation, and Signal Transduction in Activated Colon Fibroblasts

In order to identify the role of ZNF281 in activated fibroblasts, differentially expressed genes (DEGs) between CCD-18Co cells exposed to TGFβ1 or not after ZNF281 silencing were determined by RNA-sequencing. A total of 2478 DEGs were found between siZNF281 and siCtl in TGFβ1-treated-cells, 2522 DEGs between siZNF281 and siCtl in control cells, and 1712 DEGs between siCtl and siCtl TGFβ1 cells (adj *p*-value < 0.05, |log2FC| > 1.0; Appendix A). Further, to explore the biological functions of DEGs, GO term enrichment analysis was performed using DAVID [23]. Comparing enriched GO terms in the Biological Process (BP) category, we found that GO:0009966~”regulation of signal transduction”, GO:0045597~”positive regulation of cell differentiation”, GO:0016477~”cell migration”, and GO:0048468~”cell development” were among the top-10 most-enriched GO terms in both DEGs between siCtl and siCtl TGFβ1 cells (Appendix A) and DEGs between siZNF281TGFβ1 and siCtl TGFβ1 (Appendix A), suggesting that during fibroblast activation ZNF281 may mediate the effect of TGFβ1 on specific pathways. Moreover, comparing GO terms of DEGs between siZNF281 TGFβ1 and siCtl TGFβ1 and DEGs between siZNF281 and siCtl (Appendix A), we found that the top-scored GO terms were mostly overlapping, pinpointing that genes mainly involved in signal transduction and cell migration/motility are transcriptionally controlled by ZNF281 in both unstimulated and activated colon fibroblasts.

### 2.4. TGFβ1 Regulates the Expression of Fibrosis-Related Genes via ZNF281

Next, genes modulated by TGFβ1 via ZNF281 were investigated. Hence, an overlap was made between all genes regulated by TGFβ1 alone (siCtl vs. siCtl TGFβ; 1712 genes) and genes modulated by ZNF281 silencing (siCtl vs. siZNF281; 2522 genes). The results showed 507 genes significantly modulated by both TGFβ1 and ZNF281 (adj *p*-value < 0.05, |log2FC| > 1.0) (Figure 4A–C). Supervised hierarchical clustering was run on these 507 common genes, resulting in three groups of genes: Group 1 and Group 3 included genes on which ZNF281 silencing and TGFβ1 treatment showed the same effect; differently, Group 2 included genes on which ZNF281 silencing and TGFβ1 treatment had opposite effects (Figure 4D). Indeed, the modulation of Group 2 gene expression in the samples of siCtl TGFβ1 was totally or partially abolished when ZNF281 was silenced (siZNF281 TGFβ1). This evidence suggested that Group 2 genes are regulated by TGFβ1 via ZNF281. By analyzing Group 2 GO terms, it was shown that genes are mainly involved in cell differentiation and cell migration (Figure 4E, Appendix A), while the genes of Groups 1 and 3 are not (Appendix A).

### 2.5. ZNF281 Acts as a Transcriptional Activator of Pro-Fibrotic Genes Involved in Myofibroblast Differentiation

Among the 173 genes regulated by TGFβ1 via ZNF281 belonging to Group 2, 102 genes were upregulated and 71 were downregulated by TGFβ1 treatment (Appendix A). Upregulation of 102 genes was suppressed when ZNF281 was silenced. Notably, GO term enrichment analysis showed that these genes are associated with the BP and Cellular Component (CC) categories related to pathways implicated in fibroblast activation and differentiation into myofibroblasts, such as muscle cell differentiation (smooth muscle cells share several characteristics with myofibroblasts), cell migration and contractile fiber, and integrin complex, respectively (Figure 5A, Appendix A). Coherently, biological pathway enrichment using Reactome [24] highlighted that genes upregulated by TGFβ1 via ZNF281 belong to pathways strictly associated with fibrosis, such as smooth muscle contraction, ECM organization, and integrin cell surface interaction (Figure 5B). Indeed, myofibroblast markers (e.g., ACTA2, CCN2, FGF), ECM components (e.g., COL1A1, COL4A1, THBS1), remodeling enzymes (e.g., ADAM19, SERPINE1), and focal adhesion (ITGA1, ITGA11, ITGA9, ITGB3, TNS1) and contractile proteins (e.g., MYH11, MYOZ1, TPM1) are amongst the genes upregulated by TGFβ1 via ZNF281 (Appendix A).

Further, the application of STRING [25] to build the corresponding protein–protein interaction (PPI) networks showed that many proteins encoded by the 102 genes increased by TGFβ1 exposure are functionally and/or physically associated in a network (Figure 5C).

Finally, the downregulation of 71 genes by TGFβ1 was totally or partially suppressed by ZNF281 silencing. GO term enrichment analysis did not show a significant association with any BP or CC GO terms (data not shown). Moreover, no relevant PPI network or enrichment in the Reactome database was found, indicating that these genes do not belong to any pathway directly associated with fibrosis (Appendix A).

A subset of genes strongly involved in fibrogenesis was selected amongst the genes upregulated by TGFβ1 via ZNF281 (Appendix A) and analyzed by RT-qPCR (Figure 6). The results confirmed that ZNF281 mediates the transcriptional effects of TGFβ1 on: SERPINE1/PAI-1, a member of the serine proteinase inhibitor (serpin) superfamily that was recently cited as a key molecule of intestinal fibrosis [26]; THBS1 (thrombospondin-1), a matricellular glycoprotein that activates TGFβ1, thus enhancing fibrosis and compromising organ function [27]; CCN2, also known as CTGF, the connective-tissue growth factor playing a pivotal role in the pathophysiology of many fibrotic disorders [28,29]; ADAM19, an endopeptidase that cleaves extracellular matrix proteins [30]; and ITGB3, an integrin that is positively correlated to fibrosis degree [31].

Given the fundamental role of collagens in the development and progression of fibrosis, expression of COL1A1 and COL4A1, coding for type I and type IV collagen involved in ECM formation, were additionally analyzed by RT-qPCR. Results confirmed that their expression levels are controlled by TGFβ1 via ZNF281 (Figure 7A). Moreover, COL4A1 protein secretion, evaluated in the supernatant of colon fibroblasts exposed to TGFβ1 or not, after ZNF281 silencing strengthened this evidence (Figure 7B). These findings demonstrate that in colon fibroblasts, ZNF281 is required for TGFβ1-mediated upregulation of genes involved in myofibroblast differentiation, ECM composition and remodeling, and cell contraction.

## 3. Discussion

Intestinal fibrosis is a common complication of IBD and represents a great challenge for clinicians and scientists [8]. It is still unknown which factors trigger intestinal fibrosis; therefore, investigating novel molecules underlying fibrogenesis is a necessary step towards gaining effective control of fibrotic onset, progression, and cure. The currently accepted hypothesis suggests that a chronic inflammatory state activates resident fibroblasts and myofibroblasts, which are recruited to the sites of inflammation to induce wound healing. Activated fibroblasts, under the stimulus of growth factors such as TGFβ, may differentiate into myofibroblasts, which are the key players in fibrosis, being characterized by contractile and migratory as well as ECM synthesis abilities. Although to date, no therapies have been approved for intestinal fibrosis, the antifibrotic effects of different molecules that suppress fibroblast activation and differentiation into myofibroblasts have been proven in vitro and in vivo in mouse models mimicking colitis [22,32,33,34].

Therefore, investigating molecular mechanisms controlling colon fibroblast activation and differentiation into myofibroblasts is mandatory to deeply understand the pathogenesis of fibrosis as well as to identify new therapeutic targets to treat fibrosis.

Recent evidence highlights that the transcription factor ZNF281, traditionally involved in EMT regulation, is a novel player of intestinal inflammation, and its expression correlates with the disease severity degree of CD and UC patients. Very preliminary data suggest a possible implication of ZNF281 in the fibrotic process as well [21]. Since mechanisms underlying fibrogenesis in chronic colitis are largely unknown, we aimed to deeper explore this issue.

Our study underscores for the first time the role of ZNF281 in the development of gut fibrosis. Indeed, we observed a marked increase of ZNF281 in in vivo murine fibrotic colon as well as in in vitro human colon fibroblasts activated by TGFβ1 treatment. Taking advantage of a loss-of-function approach, we identified and analyzed the genes regulated by ZNF281 during colon fibroblast activation by RNA-Seq. GO term enrichment analysis showed that abrogation of ZNF281 in TGFβ1-treated fibroblasts affects the expression of genes belonging to specific pathways linked to fibrosis, such as signal transduction, regulation of cell differentiation, cell migration, and cell development. In agreement with previous studies [18,19,20] in colon fibroblasts, silencing ZNF281 determined both transcriptional activation and repression. Nevertheless, we showed that ZNF281 is a transcriptional activator downstream of TGFβ1, acting on a specific subset of genes that belong to pathways strictly associated with fibroblast activation and myofibroblast differentiation, such as smooth muscle contraction, ECM organization, and integrin cell surface interaction.

Overall, we demonstrated that ZNF281 is a key regulator of colon fibroblast activation and myofibroblast differentiation upon fibrotic stimuli by transcriptionally controlling ECM composition, remodeling, and cell contraction, highlighting a new role in gut fibrosis.

ZNF281 is already known to act as a regulator in several fundamental biological processes, such as cell differentiation and migration in other cellular contexts. For example, it has been shown that ZNF281/Zfp281 is downregulated during epithelial, muscle, and granulocytic differentiation in vitro [35], and its ectopic expression inhibits the neuronal differentiation of murine cortical neurons and neuroblastoma cells, whereas its silencing causes the opposite effect [19]. Moreover, high levels of ZNF-281 correlate with higher metastatic potential and invasiveness in several types of cancer. More specifically, ZNF281 overexpression has been observed in CRC tissues and is responsible for enhanced cell proliferation, migration, and invasion [36,37]. Additionally, ZNF281 promotes cell growth and invasion in pancreatic cancer [38]. Taken together, these results, identifying ZNF281 as a controller of relevant and strategic biological processes, underline that a dysregulation in the expression and function of the protein can be preparatory to the onset of a pathological condition, whether of a fibrotic or cancer nature.

ZNF281, or its murine orthologue Zfp281, functions in transcriptional activation and repression by binding to DNA in association with histone-modifying enzymes, such as HDAC2, PRC2, and MLL [39,40,41]. To gain insights into molecular mechanisms by which ZNF281 controls colon fibroblast activation, further efforts are required to identify genome-wide ZNF281 direct targets in activated colon fibroblasts and to characterize the chromatin landscape at ZNF281-bound promoters. TGFβ, the master regulator driving fibrosis in all organs, including the gut, acts as an upstream molecule to activate downstream signaling pathways. Canonical TGFβ signaling via Smads has a central role in the progression of fibrosis since it regulates myofibroblast proliferation as well as fibroblast transition to myofibroblasts [42]. However, non-canonical (non-Smad2/3) pathways have been reported for TGFβ and implicated in the pathogenesis of fibrosis as well [43]. We suggest that ZNF281 can act downstream of TGFβ, upregulating pro-fibrotic genes and promoting fibroblast activation, thus contributing to the onset and progression of fibrosis. Future work will be aimed at understanding if ZNF281 is induced through canonical or alternative TGFβ pathways.

Currently, there is no effective medicine to manage intestinal fibrosis, and IBD patients often undergo surgery. Identification of novel molecules involved in intestinal fibrosis may open new perspectives to clarify molecular mechanisms underlying fibrogenesis as well as target and block specific fibrogenic pathways and, eventually, allow the development of treatment methods customized to each patient’s type and degree of intestinal fibrosis. In this study, we show for the first time that the transcription factor ZNF281 is involved in TGFβ-induced fibrosis in the gut. Although the mechanism involved is still unclear, we believe that inhibiting ZNF281 could represent a therapeutic strategy to alleviate fibrosis by impairing fibroblast activation and collagen deposition.

## 4. Methods

### 4.1. Ethic Statement

Experimental procedures were previously approved by the Ministry of Health, and the study was carried out in accordance with the Italian regulations on animal welfare. The protocol was approved by the Committee on the Ethics of Animal Experiments of the Italian National Agency for New Technology, Energy and Sustainable Economic Development (ENEA) (Permit Number: 76/2017-PR, obtained on 25 January 2017).

### 4.2. Chronic DSS-Induced Colitis Model

C57BL/6 male mice (8 to 9 weeks of age) were purchased from Envigo (Inotiv Inc., Chicago, IL, USA) and were housed in collective cages at 22+/−1 °C under a 12-h light/dark cycle, with food and water provided ad libitum. Chronic DSS-induced colitis was carried out by administering 2.5% Dextran Sodium Sulpate (DSS, molecular mass, 36,000–50,000 Da, MP Biomedicals, Santa Ana, CA, USA) dissolved in autoclaved drinking water for 1 week, followed by 2 weeks of DSS disposal. This cycle was repeated 3 times. Mice were checked daily for behavior, body weight, and stool blood and consistency and were euthanized 6 and 9 weeks after their first exposure to DSS to show manifestations of early and late fibrosis, respectively [22]. At the sacrifice, colons were removed and examined for weight and length (measured from the anus to the top of the cecum). Distal colonic specimens were frozen in liquid nitrogen or fixed immediately for further analysis.

For histological analysis, samples were fixed in a 10% (*w*/*v*) formalin solution and embedded in paraffin, sectioned (4 µm thickness), and mounted on glass slides. Slices were stained using standard hematoxylin and eosin (H&E) techniques. Intestinal fibrosis was assessed by a blinded observer on adjacent sections of the distal colon stained with Masson’s Trichrome (stains collagen and proteoglycans, Bio-Optica, Milan, Italy), with scoring criteria described previously [44].

### 4.3. Immunohistochemistry

Paraffin-embedded intestinal mucosa sections (4 μm) of DSS-treated mice or controls were prepared following standard protocol. Briefly, sections were incubated in a heat-mediated antigen retrieval solution pH 6.0 (Abcam) for 10 min at 95 °C and washed in water for 5 min. Peroxidases were then inhibited by incubations in 3% H_2_O_2_ for 10 min, and sections were treated with 5% bovine serum albumin (Santa Cruz Biotech, Dallas, TX, USA) for 20 min, incubated with primary anti-ZNF antibody (TA351969, Origene, Rockville, ML, USA), and diluted 1:50 in phosphate-buffered saline 5% BSA overnight at 4 °C in a moist chamber. After washing, sections were incubated with the secondary anti-rabbit biotinylated antibody (Dako North America, Carpinteria, CA, USA) for 30 min at room temperature. The DAB detection kit (Dako) was used to visualize the antigen. Finally, sections were stained with hematoxylin and eosin.

### 4.4. Cell Culture, Treatment, and Transfection

CCD-18Co fibroblasts from normal colon were obtained from the American Type Culture Collection (ATCC) and grown in Minimum Essential Medium (MEM, Sigma, Tokyo, Japan) supplemented with 10% (*v*/*v*) fetal bovine serum (FBS), 1 mM sodium pyruvate, 2 mM L-glutamine, and penicillin–streptomycin.

For fibroblast activation, CCD-18Co were starved overnight in MEM supplemented with 0.5% (*v*/*v*) FBS, 1 mM sodium pyruvate, 2 mM L-glutamine, and penicillin–streptomycin and then treated for 24 h with 10 ng/mL TGFβ1 (Cell Signaling Technology, Danvers, MA, USA).

ZNF281 silencing was performed by transfecting CCD-18Co cells with 10 nM of Silencer pre-designed siRNAs (Ambion, Austin, TX, USA) targeting ZNF281 (siZNF281) or Silencer Negative Control (siCtl) using Lipofectamine RNAiMAX Reagent (Life Technologies, Carlsbad, CA, USA), according to the manufacturer’s instructions, for 48 h. The day after transfection, cells were starved overnight in low-serum MEM and subsequently treated for 24 h with 10 ng/mL TGFβ1.

### 4.5. RNA Isolation and RT-qPCR

Total RNA was isolated from mouse colonic tissues and CCD-18Co cells using a Direct-zol™ RNA MiniPrep Kit (Zymo Research) and reverse transcribed using an IScriptTM cDNA Synthesis Kit (BioRad, Hercules, CA, USA). The qPCR amplifications were obtained by a BioRad CFX96 TouchTM Real-Time PCR Detection System using SsoAdvanced Universal SYBR Green Super Mix (BioRad).

Primers were as depicted in the Appendix A. The expression level of each mRNA was assessed using the 2^−ΔΔCt^ method, and RLP32 and GAPDH were used for the normalization of mouse and human mRNAs, respectively.

### 4.6. Protein Isolation and Western Blot

Mouse colonic tissues as well as CCD-18Co cells were lysed in ice-cold lysis buffer (50 mM Tris (pH 7.4), 5 mM EDTA, 250 mM NaCl, 0.1% Triton X-100, 1 mM phenylmethylsulfonyl fluoride, 5 mg/mL aprotinin, 5 mg/mL leupeptin, and 1 mM sodium orthovanadate), homogenized, and incubated in ice for 30 min. Samples were centrifuged at 14,000 r.p.m. for 10 min, and supernatants were collected and analyzed by Western blot.

CCD-18Co culture supernatants were collected, supplemented with protease inhibitors (Sigma), and centrifuged at 14,000 r.p.m. for 10 min. The COL4A1 secretion was analyzed in cell supernatants by Western blot and normalized on total secreted proteins, as assessed by Coomassie Brilliant Blue staining.

For Western blot analyses, the following antibodies were used: anti-ZNF281 (ab101318, Abcam), HSP70 (H-5147, Sigma), anti-β Tubulin (T7816, Sigma), and anti-COL4A1 (PA5-86127, Life Technologies). Densitometric analyses were performed using the software ImageQuant (GE Healthcare, Life Science).

### 4.7. RNA-Sequencing

RNA-seq analysis was performed at Genomix4Life S.r.l. (Italy). TruSeq Stranded Total RNA protocol was followed after Ribo-Zero Gold treatment. Single End (75 bp) sequencing was performed on NextSeq500, aiming for 20 million reads per sample.

### 4.8. Bioinformatic Analysis

Raw fastq files were quality-checked using FastQC 0.11.2 [45]. Reads were aligned to the human genome (hg38) with TopHat v 2.1.1 [46], providing the GENCODE [47] v31 gtf annotation, achieving an overall mapping rate of 97%. Counts were summarized at gene level (GENCODE v31 annotation) using htseq-count v 0.12.4 [48], with options -f bam -s reverse-m intersection-strict-secondary-alignments = ignore-supplementary-alignments = ignore′.

Raw counts were analyzed in R using the DESeq2 package [49]. Differentially expressed genes (DEGs) were defined with a cut-off of *p* < 0.05, which was adjusted using the method of Benjamini and Hochberg using afalse discovery rate (FDR) and log2 fold-change (FC) > 1 or < −1. Hierarchical clustering and heatmaps were performed using the pheatmap package [50]. Gene Ontology (GO) term analysis was performed on DAVID 2021 [23,51], selecting the GO categories Biological Process (BP) 5, Cell Component (CC) 5, and Molecular Function (MF) 5.

Raw data in fastq format as well as un-normalized counts summarized at gene level are available through GEO Accession GSE204868.

The interaction network among proteins, encoded by candidate up- and downregulated DEGs, was achieved by using STRING database v11.5 [25] and calculating it online (https://string-db.org (accessed on 24 May 2022)) using default parameters.

Pathway enrichment of candidate up- and downregulated DEGs was analyzed using the online database Reactome [26] (https://reactome.org (accessed on 25 May 2022)).

## 5. Conclusions

Currently, the specific molecular mechanisms regulating gut fibrosis and fibroblast activation in IBD remain poorly understood, and, therefore, fully established targeted therapeutic strategies to prevent or reverse intestinal fibrosis in IBD are still lacking. Shedding more light on these mechanisms is of fundamental importance for the prevention as well as treatment of IBD-related fibrosis.

Here, we identify the transcription factor ZNF281/Zfp281 as a marker of fibrosis progression in a mouse model of inflammatory-derived gut fibrosis. Moreover, we show that during colon fibroblast activation upon TGFβ stimuli, ZNF281 is required to upregulate the expression of genes involved in ECM composition, remodeling, and cell contraction.

In conclusion, our data pinpoint that ZNF281 is a novel key regulator of colon fibroblast activation and myofibroblast differentiation by mediating the transcriptional effects of TGFβ of ECM components and remodeling enzymes as well as contraction proteins. Further effort will be required to dissect the molecular mechanisms underlying this new role for ZNF281 in gut fibrosis and to assess its potential as a therapeutic target for fibrosis pathologies that accompany chronic inflammatory diseases.

## Figures and Tables

**Figure 1 ijms-23-10261-f001:**
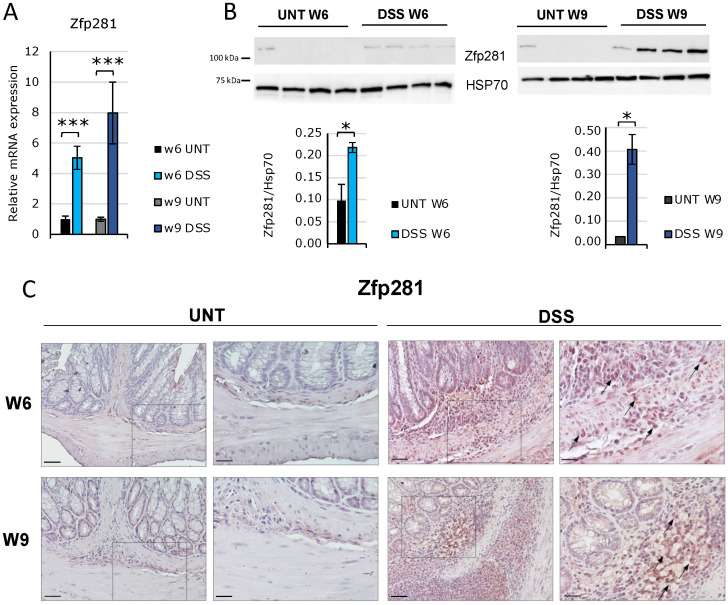
Zfp281/ZNF281 expression increased during intestinal fibrosis progression. C57BL/6 male mice were subjected to cycles of 2.5% DSS (dextran sulfate sodium) for 2 weeks, followed by 1 week of recuperation to establish a mouse intestinal fibrosis model. DSS-treated mice were sacrificed at 6 (DSS W6; *n* = 8) and 9 weeks (DSS W9; *n* = 5) of treatment along with untreated controls (UNT W6 *n* = 8; UNT W9 *n* = 8). Zfp281 mRNA (**A**) and protein levels (**B**) were analyzed in distal colon specimens. Data are expressed as mean ± SEM. * = *p*-value ≤ 0.05; *** = *p*-value ≤ 0.001. Representative photomicrographs from 4 μm colonic mucosa sections incubated with an anti-Zfp281 antibody at 20× (left panel) and 40× (right panel) magnification. The box underlines the area of the section with the highest magnification. Scale bars correspond to 20 and 10 μm, respectively. Arrows indicate Zfp281-positive inflammatory and mesenchymal cells (**C**).

**Figure 2 ijms-23-10261-f002:**
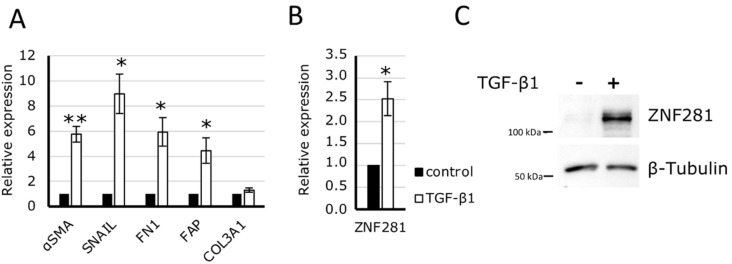
ZNF281 expression increased in the human intestinal fibroblast cell line CCD-18Co, activated by TGFβ1. CCD-18Co cells were treated with 10 ng/mL of TGFβ1for 24 h. Fibroblast activation was assessed through the dosage of several pro-fibrotic mRNAs by RT-qPCR (**A**). Inactivated colon fibroblast, ZNF281 expression was increased at both mRNA (**B**) and protein (**C**) levels. Data are expressed as mean ± SEM. * = *p*-value ≤ 0.05; ** = *p*-value ≤ 0.01 *n* = 3.

**Figure 3 ijms-23-10261-f003:**
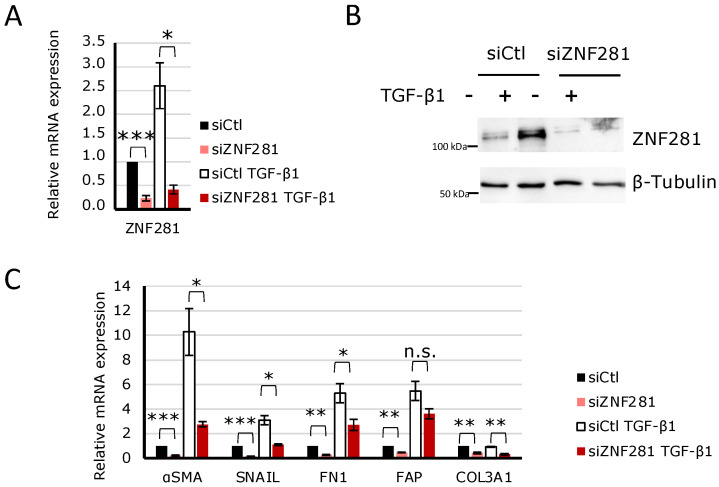
ZNF281 was required to stimulate αSMA, SNAIL, and FN1 expression in TGFβ1-activated colon fibroblasts. CCD-18Co cells were transfected with a siRNA targeting ZNF281 (siZNF281) or a negative control siRNA (siCtl). One day after transfection, cells were treated or not with 10 ng/mL of TGFβ1 for 24 h (siZNF281 TGFβ1; siCtl TGFβ1). ZNF281 silencing was verified at mRNA (**A**) and protein levels (**B**). Expression of pro-fibrotic markers was assessed by RT-qPCR (**C**). Data are expressed as mean ± SEM. * = *p*-value ≤ 0.05; ** = *p*-value ≤ 0.01; *** = *p*-value ≤ 0.001 *n* = 3, n.s. = non significant.

**Figure 4 ijms-23-10261-f004:**
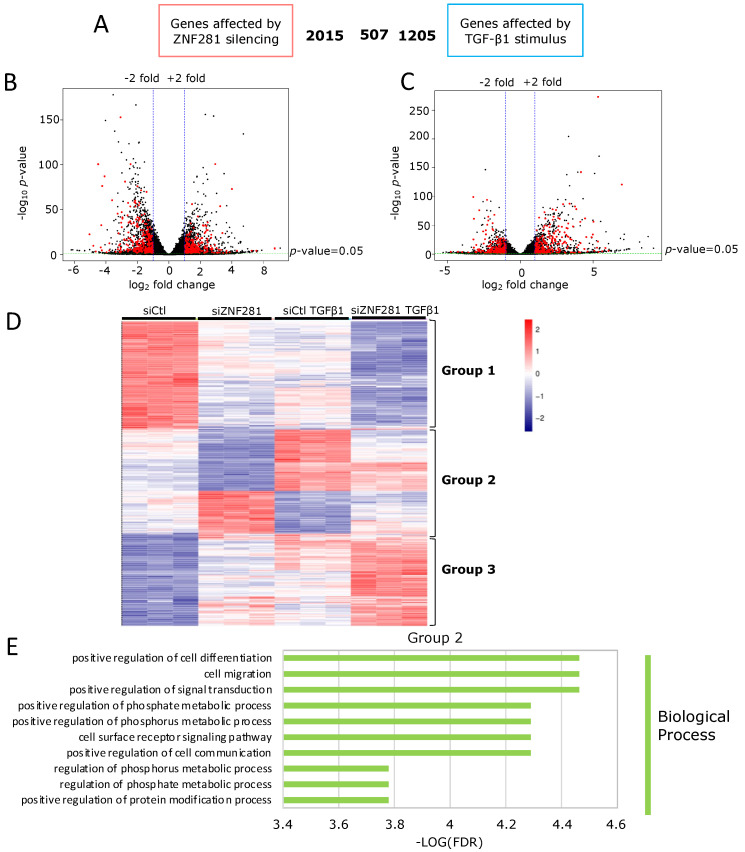
TGFβ1 regulated the expression of fibrosis-related genes, mainly involved in cell differentiation and cell migration, via ZNF281.Venn diagram depicting DEGs, as assessed by RNA-seq after ZNF281 silencing or TGFβ1 stimulus and their intersection (**A**). Volcano plot depicting differential gene expression, as assessed by RNA-seq after ZNF281 silencing (**B**) or after TGFβ1 stimulus (**C**); the 507 common genes are highlighted in red. Heatmap depicting expression of the 507 DEGs whose expression is affected by both ZNF281 silencing and by TGFβ1 stimulus (**D**). Top-enriched GO terms in the Biological Process category in genes belonging to Group 2 (**E**).

**Figure 5 ijms-23-10261-f005:**
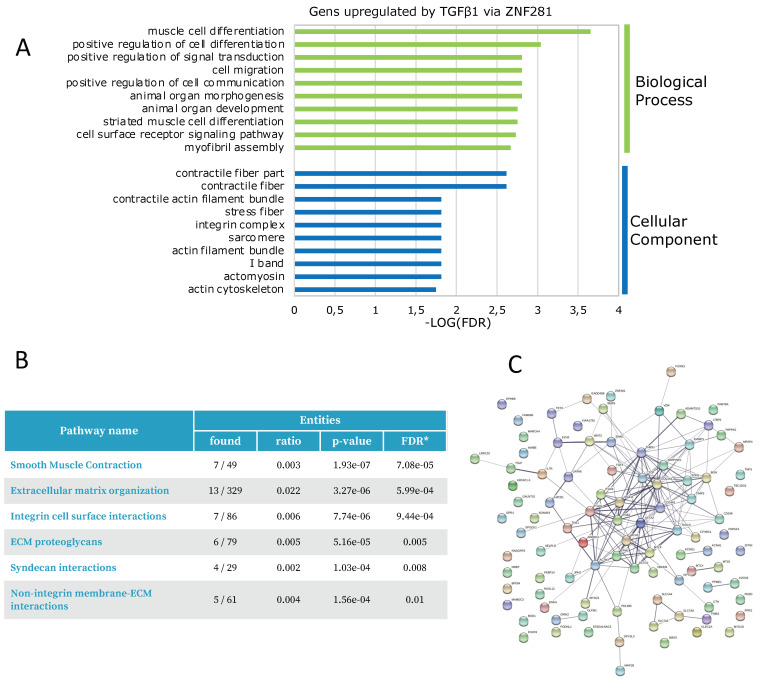
In TGFβ1-activated colon fibroblasts, ZNF281 controlled the upregulation of genes involved in myofibroblast differentiation, cell migration, contractile fibers, and integrin complexes. Functional analysis of 102 genes from Group 2, upregulated by TGFβ1 via ZNF281, was performed. Top-enriched GO terms in the Biological Process (BP) and Cellular Component (CC) categories (**A**). Reactome Biological pathway enrichment (**B**). STRING protein–protein interaction (PPI) network. The edges indicate both functional and physical protein associations (**C**).

**Figure 6 ijms-23-10261-f006:**
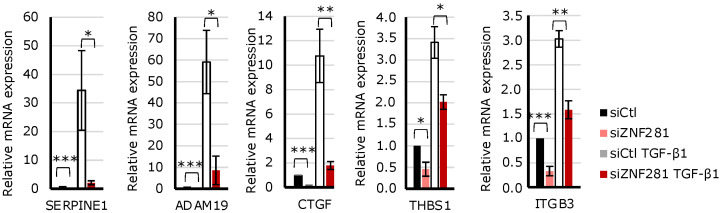
SERPINE1, ADAM19, CTGF, THBS1, and ITGB3 were upregulated by TGFβ1 via ZNF281 in colon fibroblasts. Expression of pro-fibrotic markers selected from RNA-seq data analysis was assessed by RT-qPCR in siCtl, siCtl TGFβ1, siZNF281, and siZNF281 TGFβ1 CCD-18Co cells. Data are expressed as mean ± SEM. * = *p*-value ≤ 0.05; ** = *p*-value ≤ 0.01; *** = *p*-value ≤ 0.001 *n* = 3.

**Figure 7 ijms-23-10261-f007:**
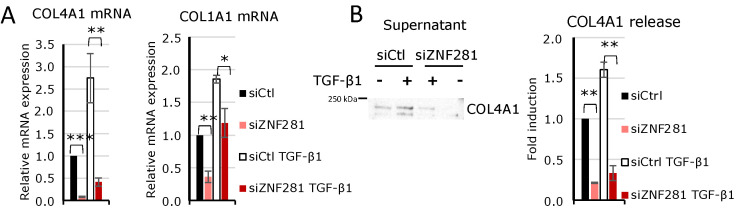
In TGFβ1-activated colon fibroblasts, ZNF281 controlled the expression of collagens COL4A1 and COL1A1 and COL4A1 secretion. Expression of COL4A1 and COL1A1 mRNAs was assessed by RT-qPCR in siCtl, siCtl TGFβ1, siZNF281, and siZNF281 TGFβ1 CCD-18Co (**A**). COL4A1 secretion was analyzed in cell supernatants by Western blot and normalized on total secreted proteins. Densitometric analysis was performed (**B**). Data are expressed as mean ± SEM. * = *p*-value ≤ 0.05; ** = *p*-value ≤ 0.01; *** = *p*-value ≤ 0.001; *n* = 3.

## Data Availability

The datasets generated for this study can be found in the GEO repository through GEO Accession GSE204868 (https://www.ncbi.nlm.nih.gov/geo/query/acc.cgi?acc=GSE204868 (accessed on 25 May 2022)).

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
