# Peer review of "ZNF281 Promotes Colon Fibroblast Activation in TGFβ1-Induced Gut Fibrosis"

_ijms, 2022, doi:10.3390/ijms231810261_

Round 1

Reviewer 1 Report

Are there previous studies on the role of ZNF281 or other Zn finger proteins in the IBD or other gastrointestinal diseases. If yes – they should be described in the introduction. If not such clear information should be added.

Mechanisms of participation of ZNF281 in fibrogenesis (if it is known) should be briefly described in the introduction

Innovative aspects of the research should be highlighted at the end of the introduction

Scale bare should be added into microphotographs instead of magnification

The micrographs could be of better quality

Whether the research had any limitations. They should be clearly discussed in the discussion chapter

Conclusion chapter with clear summarization of the research would add value to the article

The date of receipt of ethical consent should be added

Author Response

  • Are there previous studies on the role of ZNF281 or other Zn finger proteins in the IBD or other gastrointestinal diseases. If yes – they should be described in the introduction. If not such clear information should be added.

In a previous study published by the same group of authors, (PMID: 30619271), ZNF281 has been shown to be involved in intestinal inflammation, as also indicated by its increased expression levels in the inflamed colon of IBD patients. This point has been better explained in the revised manuscript in lines 69-70

  • Mechanisms of participation of ZNF281 in fibrogenesis (if it is known) should be briefly described in the introduction

In our previous paper, (PMID: 30619271), we anticipated a possible implication of ZNF281 in gut fibrosis, however, results were limited and not exhaustive, as mentioned in the introduction of the present paper. To our knowledge, there are no studies showing a role of ZNF281 in the induction of fibrosis in the gut or other organs. Our group is currently investigating the signaling and mechanisms of ZNF281 during fibrosis but results will be the object of a further paper.

  • Innovative aspects of the research should be highlighted at the end of the introduction

Following the Reviewer’s suggestion, we have better explained the innovation of our results at the end of the Introduction section (lines 73-79)

  • Scale bare should be added into microphotographs instead of magnification

Scale bars have been added to Fig 1C, Supplementary fig 1D, Supplementary Figure 2A.

  • The micrographs could be of better quality

Figures have been replaced with better quality ones.

  • Whether the research had any limitations. They should be clearly discussed in the discussion chapter

Following the Reviewer suggestion, we added a short paragraph in the Discussion section about the lack in the present paper of a description of ZNF281 mechanisms of action and the need to investigate them in the next future, such as mentioning, as an example, the mapping of DNA binding sites in activate colon fibroblasts and the co-occurrence of epigenetic modifications (lines 325-329). However, in the submitted manuscript, we have already discussed that the pathway by which TGFβ1 controls ZNF281 expression should deserve major attention in (lines 331-339)

  • Conclusion chapter with clear summarization of the research would add value to the article

A Conclusions section has been added at the end of the manuscript.

  • The date of receipt of ethical consent should be added

The date of receipt of ethical consent has been added in the Method section (line 356)

Reviewer 2 Report

Comments: Manuscript is very promising and smart work. Few issues should to be studied to clarify such this interesting work.

1-    Few English Typos errors should to be revised over all the text and caption of each figure should be rearranged.

2-    The mechanism by which DSS induced gut fibrosis was not clearly shown and If DSS is more specific for such this model or it can produce fibrosis in other organs. Authors should to clarify this point by using histopathological evidence (if possible)

3-    In section "2.2 ZNF281". pSMAD2/3 should have been successfully investigated by western blot to provide clear understand for completing signalling pathway of TGF beta. It suggested strongly to study the mechanism process of TGFβ1

4-    It is still not clear if ZNF281  can produce autocrine stimulation of the TGFβ1 pathway.

5-    REFs should have been revised and re arranged according to format of IJMS/MDPI journal.

6-    Scheme to illustrate the molecular signaling pathways between TGFβ1 and  ZNF281

Author Response

Comments: Manuscript is very promising and smart work. Few issues should to be studied to clarify such this interesting work.

  • Few English Typos errors should to be revised over all the text and caption of each figure should be rearranged. 

English typos have been fixed and figure rearranged.

  • The mechanism by which DSS induced gut fibrosis was not clearly shown and If DSS is more specific for such this model or it can produce fibrosis in other organs. Authors should to clarify this point by using histopathological evidence (if possible) 

Chronic DSS-induced colitis model has been previously set up and deeply described by others (ref n 23 in the manuscript, DOI: 10.1053/j.gastro.2017.06.013). Nevertheless, in Supplementary figure 1 and 2 we reported experimental evidence that oral administration of DSS, as described in the Method section, is sufficient to induce a chronic intestinal inflammation resulting in gut fibrosis. Macroscopic parameters were used to verify the progression of chronic intestinal inflammation and revealed a time-dependent decreased of colon length of DSS-treated mice as compared to untreated mice (Supplementary Figure 1A, B). Evaluation of body weight of each animal during the treatment was used to verify that DSS alters the intestinal barrier and provokes intestinal malabsorption with subsequent mice weight loss. Results shown that only at 9 weeks, DSS-treated mice displayed a significant weight loss due to the induced severe gut fibrosis as compared to untreated mice (Supplementary Figure 1C). Moreover, Hematoxylin and Eosin (Supplementary Figure 1D) and Masson staining on murine colon sections confirmed that DSS successfully induced chronic intestinal fibrosis. Indeed, collagen deposition in the DSS-treated mice groups was higher as compared to the control group and positive correlated with the degree of severity of gut fibrosis (Supplementary Figure 2A, B). Finally, intestinal fibrotic phenotype was further confirmed by the up-regulation of genes involved in fibrotic process both at early and late time (Supplementary 2C).

  • In section "2.2 ZNF281". pSMAD2/3 should have been successfully investigated by western blot to provide clear understand for completing signalling pathway of TGF beta. It suggested strongly to study the mechanism process of TGFβ1

We thank the Reviewer for this suggestion. We believe that dissecting this point might strongly improve our comprehension of fibrosis development. Indeed, the study of the mechanisms by which TGFb1 signaling stimulates ZNF281 expression is still ongoing in our lab and will be the topic of a further manuscript.

  • It is still not clear if ZNF281 can produce autocrine stimulation of the TGFβ1 pathway. 

We thank the Reviewer for this interesting consideration, but as we answered to point 3, we believe that the interplay between ZNF281 and TGFb1 is out of the scope of the present manuscript and will be described in a further article.

  • REFs should have been revised and re arranged according to format of IJMS/MDPI journal.

References have been formatted accordingly to IJMS style.

  • Scheme to illustrate the molecular signaling pathways between TGFβ1 and  ZNF281

As mentioned above, the molecular signaling pathways between TGFβ1 and  ZNF281 deserve major attention and will be the object of future studies.

Reviewer 3 Report

The manuscript is very interesting and generally well written. However, many flaws are present and must be resolved before publication. In particular:

Introduction: the introduction regarding the role of TGFB1 in fibrosis deserves to be improved since it is the main point interesting this study. In particular, it desearves to be specified that TGFB1 can induce fibrosis also in other organs including skin, liver and lung (PMID: 32006713, 26747705, 24187529). This is an important point to introduce because it further highlights the result obtained by the authors suggesting a similar regulation of ZNF281 also in pathologies that interests these organs.

Authors must add the molecular weigths in figures where western blot images are shown

IHC image quality must be improved and negative controls must be added

Figure 5C is unreadeable

Uniform figure captions

Remove figure number in the figures because it is already reported in figure caption

Uniform text format

Authors must add the product codes of all  primary antibodies used

Author Response

The manuscript is very interesting and generally well written. However, many flaws are present and must be resolved before publication. In particular:

  • Introduction: the introduction regarding the role of TGFB1 in fibrosis deserves to be improved since it is the main point interesting this study. In particular, it desearves to be specified that TGFB1 can induce fibrosis also in other organs including skin, liver and lung (PMID: 32006713, 26747705, 24187529). This is an important point to introduce because it further highlights the result obtained by the authors suggesting a similar regulation of ZNF281 also in pathologies that interests these organs.

Following the suggestion of the Reviewer, the role of TGFB1 in fibrosis has been deeply described in the revised manuscript in lines 48-52.

  • Authors must add the molecular weights in figures where western blot images are shown

MW have been added to western blot images

  • IHC image quality must be improved and negative controls must be added

IHC image have been replaced with better quality ones and negative controls have been added in Supplemenaty Fig 2D.

  • Figure 5C is unreadeable

Figures have been replaced with better quality ones.

  • Uniform figure captions

Captions have been reformatted

  • Remove figure number in the figures because it is already reported in figure caption

Figures have been replaced with no figure number reported.

  • Uniform text format

Text format has been improved

  • Authors must add the product codes of all  primary antibodies used 

Catalogue numbers of primary antibody have been added in the Method section. 

Round 2

Reviewer 2 Report

Although not all points were addressed. However manuscript is still so interesting and can be accepted  in present form 

Reviewer 3 Report

the manuscript can be accepted in the present form.